# Real-life use of tocilizumab with or without corticosteroid in hospitalized patients with moderate-to-severe COVID-19 pneumonia: A retrospective cohort study

Gianluca Russo[1]*, Angelo Solimini[1], Paola Zuccalà[2], Maria Antonella Zingaropoli[1], Anna Carraro[2], Patrizia Pasculli[1], Valentina Perri[1], Raffaella Marocco[2], Blerta Kertusha[2], Cosmo Del Borgo[2], Emanuela Del Giudice[2], Laura Fondaco[2], Tiziana Tieghi[2], Claudia D'Agostino[1], Alessandra Oliva[1], Vincenzo Vullo[1], Maria Rosa Ciardi[1], Claudio Maria Mastroianni[1‡], Miriam Lichtner[1,2‡]

1 Department of Public Health and Infectious Diseases, Policlinico Umberto 1st Hospital, Sapienza University of Rome, Rome, Italy, 2 Infectious Diseases Unit, S. Maria Goretti Hospital/Sapienza University of Rome, Latina, Italy

‡ These authors are joint senior authors on this work.
* gianluca.russo@uniroma1.it

## Abstract

### Objective

To evaluate the effectiveness of Tocilizumab (with or without corticosteroids) in a real-life context among moderate-to-severe COVID-19 patients hospitalized at the Infectious Diseases ward of two hospitals in Lazio region, Italy, during the first wave of SARS-CoV-2 pandemic.

### Method

We conducted a retrospective cohort study among moderate-to-severe COVID-19 pneumonia to assess the influence of tocilizumab (with or without corticosteroids) on: 1) primary composite outcome: risk for death/invasive mechanical ventilation/ICU-transfer at 14 days from hospital admission; 2) secondary outcome: COVID-related death only. Both outcomes were also assessed at 28 days and restricted to baseline more severe cases. We also evaluated the safety of tocilizumab.

### Results

Overall, 412 patients were recruited, being affected by mild (6.8%), moderate (66.3%) or severe (26.9%) COVID-19 at baseline. The median participant' age was 63 years, 56.5% were men, the sum of comorbidities was 1.34 (±1.44), and the median time from symptom onset to hospital admission was 7 [3–10] days. Patients were subdivided in 4 treatment groups: standard of care (SoC) only (n = 172), SoC plus corticosteroid (n = 65), SoC plus tocilizumab (n = 50), SoC plus tocilizumab and corticosteroid (n = 125). Twenty-six (6.3%) patients underwent intubation, and 37 (9%) COVID-related deaths were recorded. After

**Data Availability Statement:** All relevant data are within the manuscript and its Supporting information files (study dataset).

**Funding:** The authors received no specific funding for this work.

**Competing interests:** No competing interests to declare.

adjusting for several factors, multivariate analysis showed that tocilizumab (with or without corticosteroids) was associated to improved primary and secondary outcomes at 14 days, and at 28-days only when tocilizumab administered without corticosteroid. Among more severe cases the protective effect of tocilizumab (± corticosteroids) was observed at both time-points. No safety concerns were recorded.

## Conclusion

Although contrasting results from randomized clinical trials to date, in our experience tocilizumab was a safe and efficacious therapeutic option for patients with moderate-to-severe COVID-19 pneumonia. Its efficacy was improved by the concomitant administration of corticosteroids in patients affected by severe-COVID-19 pneumonia at baseline.

## Introduction

SARS-CoV-2 (Severe Acute Respiratory Syndrome–CoronaVirus 2) infection causes the Coronavirus Disease (COVID-19), characterized by a wide range of symptoms, from asymptomatic to life-threatening disease. The COVID-19 death rate by age group in Italy (period march 2020 to march 2021) ranges from 0.2% among 40–49 years-old to 9.3% among 70–79 years-old, 19.6% among 80–89 years-old, and 26.7% among > 90 years-old patients [1]. The pathogenesis of COVID-19 has not been completely elucidated, although disease severity seems to be the results of the combination of viral activity and an exaggerated host immune response [2, 3], with late host-driven inflammatory lung injury in life-threatening disease being possibly genetically related [4–6]. In COVID-19 pneumonia there is an accumulation of monocytes/macrophages and T-cells in the lungs with local overproduction of pro-inflammatory cytokines causing lung damages and possibly multi-organ failure related to the cytokine storm [2, 7], although the peak of serum cytokines (i.e. IL-6, IFNγ) in severe/critical COVID-19 patients was significantly lower than in other clinical conditions (i.e. sepsis, cytokine release syndrome (CRS) in the setting of chimeric antigen receptor (CAR) T-cell therapy, hyperinflammatory Acute Respiratory Distress Syndrome) [8]. Among cytokines, IL-6 seems to play a more prominent role possibly by inducing endothelial activation with a pro-thrombotic effect leading to thrombosis and immunothrombosis with microangiopathy in severe cases, mainly in lungs [9, 10]. Moreover, there is some evidence that genetic variants in the IL-6 inflammatory pathway may be associated with life-threatening disease [6] supporting the therapeutic strategy of IL-6 inhibition in severe COVID-19 cases.

Although it is more than a year that SARS-CoV-2 circulates with an unprecedented burden on health systems globally, a standardized and fully effective cure for COVID-19 pneumonia is lacking. It is likely that different treatment modalities might have different efficacies at different stages of illness and in different disease severity manifestations. Although many clinical studies performed worldwide, to date the pillars of the COVID-19 therapy are corticosteroids, to administer to patients in need of oxygen support [11], and anticoagulants, although their use as prophylaxis or therapy is not-yet well standardized [12]. To date, treatment targeting the virus have shown lack or limited efficacy [13, 14], whereas specific therapies targeting host immune response have given limited results needing larger studies [14]. Among specific immunomodulatory agents, tocilizumab, a monoclonal antibody against IL-6 receptor, has been one of the first drugs used as COVID-19

treatment. Tocilizumab has a long half-life (around 6.3 days after i.v. administration) and its use is already approved for some rheumatologic diseases (i.e. Rheumatoid Arthritis, Systemic Juvenile Idiopathic Arthritis diseases) and to mitigate the CRS in the setting of CAR-T cells therapy in hematologic patients. Although tocilizumab was early used as immunomodulatory drugs to treat COVID-19 patients, retrospective studies [15–17] and randomized controlled trials (RCTs) [18–24] performed so far have given inconclusive, sometime conflicting, results. Here we report a retrospective cohort study on patients hospitalized during the first wave of COVID-19 in two hospitals of the Lazio region, Italy. We assessed the efficacy of tocilizumab (with or without corticosteroids) in a "real-life" context among hospitalized patients with moderate-to-severe COVID-19 pneumonia.

## Material and methods

### Study design, inclusion criteria and outcomes

We conducted a retrospective cohort study using medical records of the Infectious Diseases division of Sapienza University hospitals Policlinico Umberto 1st (Rome) and S. Maria Goretti (Latina), Lazio region, Italy. We included adult patients consecutively accessing the hospitals through the Emergency Departments from March 5th to July 13rd, 2020, clinically in need of hospitalization because of fever and/or respiratory symptoms, having a positive reverse-transcriptase polymerase-chain reaction (RT-PCR) assay for SARS-CoV-2 on nasopharyngeal swab. We excluded critical COVID-19 patients -requiring immediate orotracheal intubation (OTI) for invasive mechanical ventilation (IMV) and/or transfer to ICU (Intensive Care Unit)- within 24h after hospital admission. At the time of this study, because of shortage of ICU-beds, the ICU-transfer was reserved only to patients mechanically ventilated. Following the up-to-date advices from the Italian Society of Infectious and Tropical Diseases (www.simit. org), all participants have received as standard of care (SoC) a combination of Lopinavir/Ritonavir, Hydroxychloroquine, Antibiotics (mainly azithromycin), Low-Weight Molecular Heparin (LWMH) as prophylaxis or treatment according to D-dimer values, and oxygen support (through Venturi Mask or Continuous Positive Airway Pressure -CPAP- helmet) when needed. Patients were divided in 4 treatment groups, defined as receiving SoC only or one additional therapeutic option to SoC (CCS, corticosteroid alone; TCZ, tocilizumab alone; TCZ +CCS, tocilizumab plus corticosteroid) if they were receiving one or both of them after hospital admission. Tocilizumab was administered as "off-label" internal use intravenously (8 mg/kg) or subcutaneously (324 mg) according to availability, once or twice following physician decision. The majority of patients taking CCS received a course of methylprednisolone (72%) (30–40 mg twice a day with tapering over 10–14 days), whereas the remaining received dexamethasone (28%) (8–12 mg/day with tapering over 10–14 days). Vital parameters and clinical data were collected at the hospital admission. At baseline all patients underwent blood gas analysis, blood exams, and thorax computer tomography (CT) scan. Basic blood exams evaluated for outcomes (Cells Blood Count, Ferritin, Lactate dehydrogenase, D-dimer, C-Reactive Protein -CRP) were performed at baseline and 5–7 days after hospitalization. Patients were classified for clinical severity (mild, moderate, severe) according to NIH-COVID-19 criteria updated on December 17th, 2020 (https://www.covid19treatmentguidelines.nih.gov/overview/clinical-spectrum/). The study primary outcome was death for every cause or IMV or ICU-transfer, whatever came first in the 14 days following admission. Patients who did not have the outcome on or were discharged before day 14 were censored at discharge date or day 14, whichever occurred first. As secondary outcome we considered only COVID-related deaths within 14-days from admission. We also assessed primary and secondary outcomes at day 28 from hospitalization, and we also restricted the analysis according to clinical severity

(moderate-to-severe) at baseline. The safety of TCZ was also evaluated by monitoring the occurrence of bacterial superinfections or other adverse events.

The study was approved by the Ethical Committee of both participating hospitals (Policlinico Umberto 1st/Sapienza University of Rome: protocol number 5819/2020; Lazio 2: protocol number 1960080757/2020). All patients who received tocilizumab provided written informed consent.

## Statistical analysis

Exploratory analysis was carried out by tabulating proportion for categorical variables and median with interquartile range (IQR) for continuous variables by treatment group (SoC, CCS, TCZ, TCZ+CCS). Heterogeneity between treatment groups were assessed with chi-square test of independence or Fisher's exact test for categorical response variables and with Kruskal-Wallis test for continuous response variables. In the main analysis, we calculated the hazard ratio (HR) of each treatment with using Cox proportional hazard regression models adjusted for age, gender, $PaO_2/FiO_2$ at baseline, CRP at baseline, days from symptoms onset, sum of comorbidities. Those variables were chosen because used in previous similar works. We excluded some variables because of: 1) large number of missing observations; 2) number of events too small to calculate hazard ratios; 3) high correlation with other variables already selected for the model. Each hospital was included using a clustering term. In secondary analyses we compared TCZ administrations modalities. All statistical analyses were performed using version 3.6.2 of the R programming language (R Project for Statistical Computing; R Foundation).

## Results

A total of 442 COVID-19 patients were hospitalized at the Infectious Diseases Units of Latina and Rome Hospitals, Italy: 30 patients were excluded because of incomplete outcome data (n = 23) and/or death or IMV or ICU-transfer within 24h after hospital admission (n = 7). Thus, a total of 412 COVID-19 patients equally distributed between clinical centres were included in this retrospective cohort study (Fig 1).

The baseline patients' characteristics for each treatment' group are resumed in Table 1. Overall, the median age of participants was 63 [IQR: 51–75] years, and 56.5% were men. The main comorbidities reported, with some differences between treatment' groups (Table 1), were: cardiovascular diseases (43.7%), diabetes mellitus (18.2%), chronic pulmonary disease (14.8%), cancer (8.7%), chronic renal failure (7.8%), obesity (5.3%). Patients were seeking hospital care after a median of 7 [IQR 3–10] days from symptoms onset, reporting fever (73.5%), cough (48.8%), dyspnoea (33.5%), myalgia (23.8%) as main ongoing symptoms (Table 1). Results of blood exams at the admission and 5–7 days later are reported in Table 2. At baseline patients of the TCZ+CCS group showed lower lymphocytes count and higher neutrophils/lymphocytes ratio, as well as higher ferritin, LDH, D-dimer and CRP (Table 2). The median of $PaO_2/FiO_2$ ratio at baseline, as well as its lowest value throughout hospitalization, was smaller in the TCZ+CCS group (Table 3). The vast majority of participants (n = 384, 93.2%) had CT-scan findings of bilateral interstitial pneumonia at baseline. Based on baseline clinical evaluation, CT-scan and blood gas analysis results, patients were classified as mild (6.8%), moderate (66.3%) or severe (26.9%) disease case. Throughout hospitalization one third of participants (33.2%) was not in need of oxygen support, whereas 35.4% (n = 146) received oxygen through Venturi Mask and 25% (n = 103) through CPAP helmet, and 6.3% (n = 26) was mechanically ventilated. Throughout hospitalization we observed an overall case-fatality rate of 12.4% (51 deaths), being 9% (37 deaths) COVID-related and 3.4% (14 deaths) non-COVID-related.

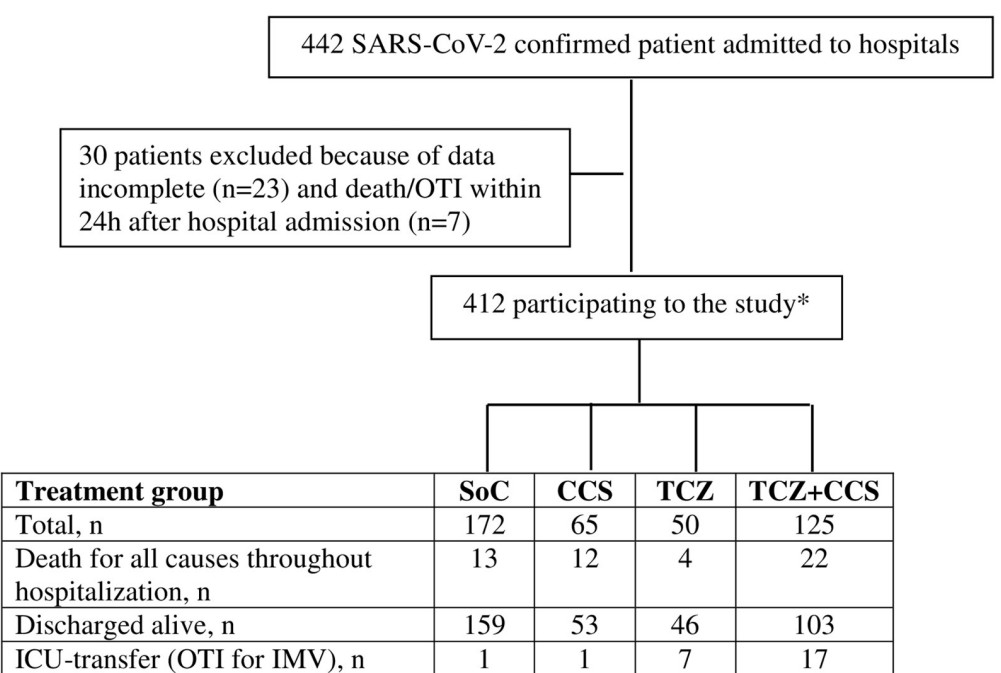

Fig 1. Population study by treatment group.

For the primary composite outcome, we registered 40 events by day-14 and 60 events throughout hospitalization, both more frequent in the TCZ+CCS group (Table 3). The majority of COVID-related deaths throughout hospitalization (n = 24/37, 64.9%) were observed in patients receiving TCZ, with corticosteroid (TCZ+CCS: n = 20/37, 54.0%) or without (TCZ: n = 4/37, 10.8%) (Table 3). Furthermore, intubation occurred more frequently in patients receiving tocilizumab: among the 26 patients receiving IMV during the hospitalization, 17 (70.8%) were in the TCZ+CCS group, 7 (26.9%) in the TCZ group and, 1 (3.8%) in SoC and CCS groups each (Table 3). Concerning non-COVID-related death (n = 14), two (14.3%) were reported in the TCZ+CCS group, none in the TCZ group, and 6 (42.8%) in SoC and CCS groups each (Table 3).

Tocilizumab was administered a median of 10 days from symptoms onset and 3 days after hospital admission, mainly intravenously (93%), once (53%) or twice (47%) according to treating physicians that were more prone to assign more severe cases at baseline to TCZ and TCZ+CCS groups (Table 3). Moreover, there was a different distribution of participants by treatment group between clinical centres, with patients from Latina Hospital more prone to receive corticosteroid therapy according to local clinical practice. The median duration of corticosteroid administration was longer in TCZ+CCS than CCS group (13 [9–18] vs 9 [6–14] days, p = 0.007) (Table 1). Concerning adverse events related to tocilizumab, we observed few cases of Grade 1–2 increase of transaminases and three cases of severe neutropenia that recovered spontaneously without consequences. Overall, we registered 71 infections among 62 patients,

**Table 1. Baseline characteristics of participants by treatment group.**

| | SoC (n = 172) | CCS (n = 65) | TCZ$^{\S}$ (n = 50) | TCZ$^{\S}$ + CCS (n = 125) | P-value[1] | Total (n = 412) |
|---|---|---|---|---|---|---|
| **Healthcare facility** | | | | | | |
| Policlinico Umberto I, Rome | 92 (53.5%) | 23 (35.4%) | 43 (86%) | 30 (24%) | <0.001 | 188 (45.6%) |
| S. Maria Goretti, Latina | 80 (46.5%) | 42 (64.6%) | 7 (14%) | 95 (76%) | | 224 (54.4%) |
| **Demographic characteristics** | | | | | | |
| Age, years, median [IQR] | 60 [47–73] | 66 [52–82] | 63 [51–75] | 64 [55–74] | 0.14 | 63 [51–75] |
| Male gender, n (%) | 88 (51%) | 29 (45%) | 29 (58%) | 87 (70%) | 0.002 | 233 (56.5%) |
| **Co-morbidities** | | | | | | |
| Sum, mean (±SD) | 1.16 (±1.30) | 1.57 (±1.17) | 1.38 (±1.66) | 1.46 (±1.64) | 0.03 | 1.34 (±1.44) |
| Charlson index, mean (±SD) | 1.58 (±1.96) | 1.67 (±2.06) | 1.55 (±1.46) | 1.11 (±1.54) | 0.068 | 1.45 (±1.81) |
| Chronic cardiovascular diseases | 65 (38%) | 34 (52%) | 24 (48%) | 57 (46%) | 0.2 | 180 (43.7%) |
| Chronic pulmonary diseases | 19 (11%) | 14 (22%) | 8 (16%) | 20 (16%) | 0.2 | 61 (14.8%) |
| Chronic renal failure | 13 (7.6%) | 6 (9.2%) | 1 (2%) | 12 (9.6%) | 0.3 | 32 (7.8%) |
| Diabetes Mellitus | 28 (16%) | 12 (18%) | 9 (18%) | 26 (21%) | 0.8 | 75 (18.2%) |
| Cancer | 7 (4.1%) | 9 (14%) | 6 (12%) | 14 (11%) | 0.022 | 36 (8.7%) |
| Obesity (BMI>30) | 4 (2.3%) | 5 (7.7%) | 1 (2%) | 12 (9.6%) | 0.022 | 22 (5.3%) |
| Dyslipidaemia | 19 (11%) | 4 (6.2%) | 6 (12%) | 14 (11%) | 0.7 | 43 (10.4%) |
| Hypothyroidism | 12 (7%) | 2 (3.1%) | 2 (4%) | 12 (9.6%) | 0.4 | 28 (6.8%) |
| Neurodegenerative diseases | 14 (8.1%) | 9 (14%) | 1 (2%) | 4 (3.2%) | 0.021 | 28 (6.8%) |
| Autoimmune disease* | 4 (2.3%) | 3 (4.6%) | 9 (18%) | 4 (3.2%) | <0.001 | 20 (4.9%) |
| Other disease** | 14 (8.1%) | 4 (6.2%) | 2 (4%) | 8 (6.4%) | 0.8 | 28 (6.8%) |
| **Clinical presentation at baseline** | | | | | | |
| Days from symptoms onset^, median [IQR] | 7 [3–12] | 5 [2–10] | 7 [4.5–8.5] | 7 [4–10] | 0.4 | 7 [3–10] |
| Fever, n (%)[a] | 115 (68%) | 39 (63%) | 44 (90%) | 105 (88%) | <0.001 | 303 (73.5%) |
| Cough, n (%) | 76 (44%) | 25 (38%) | 30 (60%) | 70 (56%) | 0.025 | 201 (48.8%) |
| Dyspnoea, n (%)[b] | 47 (28%) | 19 (31%) | 22 (46%) | 50 (42%) | 0.032 | 138 (33.5%) |
| Diarrhoea, n (%)[c] | 23 (14%) | 5 (8.1%) | 8 (16%) | 6 (5%) | 0.048 | 42 (10.2%) |
| Myalgia, n (%)[d] | 44 (26%) | 8 (13%) | 20 (41%) | 26 (22%) | 0.006 | 98 (23.8%) |
| Loss of taste/smell, n (%) | 16 (9.3%) | 2 (3.1%) | 3 (6%) | 2 (1.6%) | 0.023 | 23 (5.6%) |
| Conjunctivitis, n (%)[e] | 0 (0%) | 2 (3.3%) | 0 (0%) | 3 (2.5%) | 0.065 | 5 (1.2%) |
| PaO$_2$/FiO$_2$ ratio[f], median [IQR] | 400 | 350 | 332 | 287 | <0.001 | 350 |
| | [349–457] | [300–400] | [281–378 | [195–343] | | [285–410] |
| **Disease severity classification at baseline** | | | | | | |
| Mild | 24 (14%) | 3 (4.6%) | 1 (2%) | 0 (0%) | 0.001 | 28 (6.8%) |
| Moderate | 132 (76.7%) | 47 (72.3%) | 36 (72%) | 58 (46.4%) | | 273 (66.3%) |
| Severe | 16 (9.3%) | 15 (23.1%) | 13 (26%) | 67 (53.6%) | | 111 (26.9%) |
| **Days to TCZ and/or CCS administration** | | | | | | |
| Days to TCZ administration, median [IQR] | - | - | 3 [1–6] | 3 [1–5] | 0.5 | 3 [1–5] |
| *Missing, n (%)* | - | - | 10 (20%) | 5 (4%) | | 15 (8.6%)° |
| Days to CCS administration, median [IQR] | - | 0 [0–1] | - | 1 [0–5] | 0.015 | 1 [0–4] |
| *Missing, n (%)* | - | 13 (20%) | - | 11 (8.8%) | | 24 (12.6%)°° |

(*Continued*)

**Table 1.** (Continued)

| | SoC (n = 172) | CCS (n = 65) | TCZ[§] (n = 50) | TCZ[§] + CCS (n = 125) | P-value[1] | Total (n = 412) |
|---|---|---|---|---|---|---|
| **Duration of hospitalization** | | | | | | |
| Days, median [IQR] | 12 [9–19] | 16 [11–31] | 20 [15–30] | 30 [22–37] | <0.001 | 19 [11–31] |

SoC: Standard of Care; CCS: corticosteroid; TCZ: tocilizumab; BMI: Body Mass Index;

[§] TCZ administration (iv 8 mg/kg per dose; sc 324 mg per dose): once iv (52,6%), twice iv (40%), twice sc (7.4%);

[1] Statistical test performed: chi-square test of independence; Kruskal-Wallis test; Fisher's exact test.

[*]Autoimmune disease includes: psoriasis (n = 6), LES (n = 2), Sclerodermia (n = 2), Idiopathic arthritis (n = 1), Primary biliary cirrhosis (n = 2), Ulcerative recto-colitis (n = 2), Multiple Sclerosis (n = 2), Undifferentiated connectivity (n = 3);

[**]Other disease includes: Gastro-oesophageal reflux (n = 9), Psychiatric disorders (n = 6), Benign prostatic hypertrophy (n = 3), Seasonal allergic condition (n = 3), HCV-infection (n = 4), HIV-infection (n = 1), HBV-infection (n = 1), Brugada syndrome (n = 1).

Data available for or *n* participants (%)::

[^]356/412 (86.4%);

[a]399/412 (96.8%);

[b]397/412 (96.4%);

[c]398/412 (96.6%);

[d]399/412 (96.8%);

[e]396/412 (96.1%);

[f]389/412 (94.4%).

[°] 15/175 TCZ recipients;

[°°] 24/190 CCS recipients.

including 4 cases of *Clostridium difficile* and 5 fungal infections (4 *Candida spp* and 1 *Aspergillus spp*). The secondary infections occurred in all groups (26 in SoC, 16 in CCS, 9 in TCZ, 20 in TCZ+CCS) without differences by groups comparison (p>0.2). Non-COVID-related death was not influenced by TCZ administration (Table 3).

By comparing survivors vs non-survivors for COVID-related deaths (Table 4) we found that non-survivors were more aged, more likely affected by other diseases (chronic cardiovascular and renal diseases, obesity, neurodegenerative diseases), more likely to complain for baseline shortness of breath, with a lower respiratory function ($PaO_2/FiO_2$) at baseline and during hospitalization, leading to higher oxygen supply needs, and more likely to have worse baseline laboratory tests (Table 4). No differences between recruiting centres were observed. The length of hospital stay was longer, although not statistically significant, among non-survivors (14 vs 19 days, p = 0.2). Finally, no differences between survivors and non-survivors were observed for the timing of TCZ (with or without corticosteroid) administration (Table 4).

Results of multivariate Cox proportional hazard regression models, after adjusting for age, gender, days from symptoms onset, sum of comorbidities, healthcare centres, baseline CRP and $PaO_2/FiO_2$ ratio, are shown in Table 5. A significant reduction of the risk for the primary composite outcome at 14 days was associated to TCZ and TCZ+CCS groups, whereas at 28-days remained associated to TCZ only (Table 5). In the subgroup of patients with severe disease we found similar protective effect for the primary composite outcome in TCZ and TCZ+CCS at 14-days and 28-days from hospital admission (Table 5). The protective effects remained similar after excluding from the analysis the 11 patients who received a 5-days course of remdesivir (5 in TCZ+CCS group, 3 in CCS and 3 in SoC), or when TCZ number of doses were considered (S1 Table). The sum of comorbidities was significantly associated to higher risk for the primary composite outcome at 14 days (HR 1.23, 95%CI: 1.10–1.38,

**Table 2. Blood count and biochemical markers at baseline and 5–7 days after hospital admission by treatment' group.**

| | SoC (n = 172) | CCS (n = 65) | TCZ (n = 50) | TCZ + CCS (n = 125) | P-value[^] | Total (n = 412) |
|---|---|---|---|---|---|---|
| **Blood count and biochemical markers at baseline** | | | | | | |
| White blood cells, n/μl, median [IQR] | 5,515 | 5,830 | 4,820 | 5,850 | 0.4 | 5,600 |
| | [4,142–7,500] | [4,120–8,900] | [3,690–7,320] | [4,200–8,280] | | [4,090–7,715] |
| *Missing, n (%)* | *4 (2.3%)* | *1 (1.5%)* | *1 (2%)* | *0 (0%)* | | *7 (1.7%)* |
| Neutrophils, n/μl, median [IQR] | 3,580 | 3,650 | 3,390 | 4,385 | 0.054 | 3,680 |
| | [2,304–5,370] | [2,450–6,420] | [2,390–5,410] | [2,850–6,580] | | [2,450–5,480] |
| *Missing, n (%)* | *5 (2.9%)* | *0 (0%)* | *1 (2%)* | *1 (0.8%)* | | *7 (1.7%)* |
| Lymphocytes, n/μl, median [IQR] | 1,295 | 960 | 1,000 | 880 | <0.001 | 1,070 |
| | [895–1,730] | [730–1,680] | [640–1,390] | [650–1,280] | | [740–1,585] |
| *Missing, n (%)* | *4 (2.3%)* | *0 (0%)* | *1 (2%)* | *0 (0%)* | | *5 (1.2%)* |
| N/L ratio[a], median [IQR] | 2.8 | 3.5 | 4 | 4.4 | <0.001 | 3.4 |
| | [1.6–4.6] | [2.2–5.4] | [2–6.8] | [2.7–8.7] | | [2.0–6.0] |
| *Missing, n (%)* | *5 (2.9%)* | *0 (0%)* | *1 (2%)* | *1 (0.8%)* | | *7 (1.7%)* |
| Ferritin, μg/L, median [IQR] | 302 | 246 | 585 | 634 | <0.001 | 394 |
| | [167–498] | [159–596] | [250–1,014] | [359–1,432] | | [206–864] |
| *Missing, n (%)* | *67 (38.9%)* | *32 (49.2%)* | *15 (30%)* | *46 (36.8%)* | | *160 (38.8%)* |
| LDH[b], U/L, median [IQR] | 214 | 281 | 286 | 302 | <0.001 | 256 |
| | [176–260] | [214–336] | [227–372] | [241–390] | | [200–331] |
| *Missing, n (%)* | *17 (9.9%)* | *5 (7.7%)* | *3 (6%)* | *11 (8.8%)* | | *36 (8.7%)* |
| D-dimer, μg FEU/ml [IQR] | 0.90 | 1.04 | 1.10 | 1.56 | 0.11 | 1.09 |
| | [0.48–1.70] | [0.55–2.74] | [0.66–1.79] | [0.90–2.37] | | [0.6–2.08] |
| *Missing, n (%)* | *50 (29.1%)* | *18 (27.7%)* | *6 (12%)* | *45 (36%)* | | *119 (28.9%)* |
| CRP[c], mg/dl, median [IQR] | 1 | 3 | 5 | 6 | <0.001 | 3 |
| | [0–4] | [1–9] | [2–10] | [2–12] | | [1–8] |
| *Missing, n (%)* | *12 (7%)* | *3 (4.6%)* | *3 (6%)* | *7 (5.6%)* | | *25 (6.1%)* |
| **Blood count and biochemical markers at 5–7 days after hospital admission** | | | | | | |
| White blood cells, n/μl, median [IQR] | 5,290 | 9,020 | 4,585 | 8,205 | <0.001 | 6,170 |
| | [4,225–7,015] | [5,490–12,575] | [3,445–5,268] | [6,140–11,008] | | [4,510–9,005] |
| *Missing, n (%)* | *52 (30.2%)* | *14 (21.5%)* | *6 (12%)* | *9 (7.2%)* | | *81 (19.7%)* |
| Neutrophils, n/μl, median [IQR] | 3,025 | 6,750 | 2,680 | 7,100 | <0.001 | 4,050 |
| | [2,308–4,042] | [3,745–9,628] | [1,818–3,780] | [4,325–9,115] | | [2,620–7,275] |
| *Missing, n (%)* | *60 (34.9%)* | *17 (26.1%)* | *6 (12%)* | *10 (8%)* | | *93 (22.6%)* |
| Lymphocytes, n/μl, median [IQR] | 1,450 | 1,180 | 1,085 | 810 | <0.001 | 1,125 |
| | [1,010–1,880] | [820–1,950] | [652–1,730] | [520–1,240] | | [710–1,730] |
| *Missing, n (%)* | *59 (34.3%)* | *17 (26.1%)* | *6 (12%)* | *10 (8%)* | | *92 (22.4%)* |
| N/L ratio[a], median [IQR] | 2 | 4 | 2 | 9 | <0.001 | 3 |
| | [2–3] | [2–10] | [1–5] | [5–17] | | [2–9] |
| *Missing, n (%)* | *60 (34.9%)* | *17 (26.1%)* | *6 (12%)* | *10 (8%)* | | *93 (22.6%)* |
| Ferritin, μg/L, median [IQR] | 286 | 327 | 598 | 673 | <0.001 | 430 |
| | [129–490] | [195–913] | [409–909] | [360–1,214] | | [220–912] |
| *Missing, n (%)* | *110 (63.9%)* | *44 (67.7%)* | *21 (42%)* | *47 (37.6%)* | | *222 (53.9%)* |
| LDH[b], U/L, median [IQR] | 186 | 232 | 280 | 297 | <0.001 | 246 |
| | [164–254] | [189–316] | [214–328] | [230–366] | | [185–316] |
| *Missing, n (%)* | *90 (52.3%)* | *34 (52.3%)* | *12 (24%)* | *24 (19.2%)* | | *160 (38.8%)* |

*(Continued)*

**Table 2.** (Continued)

| | SoC (n = 172) | CCS (n = 65) | TCZ (n = 50) | TCZ + CCS (n = 125) | P-value^ | Total (n = 412) |
|---|---|---|---|---|---|---|
| D-dimer, µg FEU/ml [IQR] | 0.72 | 1.18 | 1.38 | 1.98 | <0.001 | 1.45 |
| | [0.41–0.61] | [0.61–4.14] | [1.06–2.77] | [1.16–4.47] | | [0.71–3.18] |
| *Missing, n (%)* | *94 (54.6%)* | *38 (58.5%)* | *6 (12%)* | *20 (16%)* | | *158 (38.3%)* |
| CRP[c], mg/dl, median [IQR] | 2 | 1 | 2 | 1 | 0.008 | 1 |
| | [0–6] | [0–3] | [1–6] | [0–3] | | [0–4] |
| *Missing, n (%)* | *56 (32.6%)* | *22 (33.8%)* | *5 (10%)* | *13 (10.4%)* | | *96 (23.3%)* |

SoC: Standard of Care; CCS: corticosteroid; TCZ: tocilizumab; IQR: Interquartile Range;

^ Statistical test performed: chi-square test of independence; Kruskal-Wallis test; Fisher's exact test.

[a] Neutrophil to Lymphocyte ratio;

[b] Lactate dehydrogenase;

[c] C-Reactive Protein.

p<0.001) and 28 days (HR 1.20, 95%CI: 1.10–1.32, p<0.001). Results of the analysis for the secondary outcome (COVID-related death) among all participants at 14- and 28-days after hospital admission showed a protective effect in TCZ and TCZ+CCS groups as for the primary composite outcome, but with lower hazard ratios (Table 5). In the subgroup of patients with severe disease we found a protective effect for the secondary outcome in TCZ and TCZ+CCS at 14-days and 28-days from hospital admission (Table 5). The use of corticosteroids alone seemed to be not influencing study outcomes.

## Discussion

After the first study on the use of TCZ in COVID-19 patients from China [25], many retrospective studies (S2 Table) and six Randomized Clinical Trials (RCTs) (S3 Table) have been published to date. Results of studies on TCZ effectiveness were jeopardized, mainly because of different baseline disease severity. Among moderate-to-severe COVID-19 patients at baseline, some observational studies comparing TCZ to SoC showed clinical improvement 10–14 days after admission [26], and lower [27, 28] or similar [29] in-hospital mortality. Other retrospective studies in patients with similar disease severity, but in which concomitant steroid therapy was administered (30 to 70% of participants), showed improved outcomes at 30-days [30–34]. RCTs among moderate-to-severe COVID-19 patients at baseline showed lack of clinical improvement [18, 19], lower 28-days mortality risk (HR 0.56, 95%CI: 0.33–0.97, p = 0.04) [20] or no influence on mortality (p = 0.64) [19]. Concerning severe-to-critical COVID-19 patients at baseline, retrospective studies on TCZ efficacy showed improved outcomes at various timepoints after admission [17, 35–42], being associated to critical disease only in one study [41] or to CRP>15 mg/dl at baseline only [39, 42]. RCTs among severe-to-critical COVID-19 patients at baseline showed discordant results, with one displaying lower risk for IMV or death at day-14 (median posterior HR 0.58, 90%CrI 0.33–1.00) but not at day-28 from hospital admission [21], and others achieving no clinical improvement at day-15 [22] or day-28 [24] of hospital stay. Notably, a higher 28-days mortality at the limit of statistical significance was associated to TCZ (OR: 2.7, 95%CI 0.97–8.35, p = 0.07) in one RCT that was early stopped [22]. Moreover, results of RCT REMAP-CAP comparing anti-IL-6 (plus corticosteroid) vs SoC in severe-to-critical COVID-19 cases showed higher in-hospital survival (aOR 1.64 95%CrI 1.14, 2.35; posterior probability of superiority 99.6%) and higher organ support-free days (aOR 1.64, 95%CrI 1.25, 2.14, posterior probability of superiority >99.9%) in TCZ recipients [23]. Furthermore,

**Table 3. Respiratory support and clinical outcomes by treatment' group.**

| | SoC (n = 172) | CCS (n = 65) | TCZ (n = 50) | TCZ + CCS (n = 125) | P-value[^] | Total (n = 412) |
|---|---|---|---|---|---|---|
| **$PaO_2/FiO_2$ and respiratory support throughout hospital stay** | | | | | | |
| *$PaO_2/FiO_2$ (P/F) lowest value[1]* | | | | | | |
| Median [IQR] | 363 | 283 | 169 | 147 | <0.001 | 228 |
| | [289–424] | [188–357] | [127–239] | [110–186] | | [149–364] |
| *Respiratory support administered, n (%)* | | | | | | |
| None | 113 (65.7%) | 24 (36.9%) | 0 (0%) | 0 (0%) | <0.001 | 137 (33.2%) |
| Venturi Mask | 50 (29.1%) | 29 (44.6%) | 28 (56%) | 39 (31.2%) | | 146 (35.4%) |
| CPAP helmet | 8 (4.6%) | 11 (16.9%) | 15 (30%) | 69 (55.2%) | | 103 (25%) |
| OTI | 1 (0.6%) | 1 (1.5%) | 7 (14%) | 17 (13.6%) | | 26 (6.3%) |
| **Deaths** | | | | | | |
| *COVID-related death* | | | | | | |
| *At day-14 of hospital stay* | | | | | | |
| n (%) | 7 (4.1%) | 5 (7.7%) | 1 (2%) | 6 (4.8%) | 0.5 | 19 (4.6%) |
| Days to death, median [IQR] | 6 [3.5–9] | 10 [7–11] | 6 [6–6] | 12.5 [10.5–13] | 0.027 | 10 [6–11] |
| *Throughout the hospital stay* | | | | | | |
| n (%) | 7 (4.2%) | 6 (10%) | 4 (8%) | 20 (16.0%) | 0.006 | 37 (9%) |
| Days to death, median [IQR] | 6 [4–9] | 10 [8–11] | 20 [13–30] | 23 [14–40] | <0.001 | 14 [10–29] |
| *Non COVID-related death* | | | | | | |
| *At day-14 of hospital stay* | | | | | | |
| n (%) | 3 (1.7%) | 1 (1.5%) | 0 (0%) | 1 (0.8%) | >0.9[^] | 5 (1.2%) |
| Days to death, median [IQR] | 7 [4.5–8.5] | 9 [9–9] | NA | 12 [12–12] | 0.3 | 9 [7–10] |
| *Throughout the hospital stay* | | | | | | |
| n (%) | 6 (3.5%) | 6 (9.2%) | 0 (0%) | 2 (1.6%) | 0.032 | 14 (3.4%) |
| Days to death, median [IQR] | 16 [8–28] | 34 [20–43] | NA | 22 [17–26] | 0.2 | 26 [10–31] |
| **Orotracheal intubation (OTI)** | | | | | | |
| n (%) | 1 (0.6%) | 1 (1.5%) | 7 (14%) | 17 (13.6%) | <0.001 | 26 (6.3%) |
| Days to OTI, median [IQR] | 3 [3–3] | 1 [1–1] | 3 [2–8] | 6 [2–12] | 0.4 | 5 [2–10] |
| **Composite primary outcome** | | | | | | |
| *At day-14 of hospital stay* | | | | | | |
| n (%) | 10 (5.8%) | 6 (9.2%) | 7 (14%) | 17 (13.6%) | 0.082 | 40 (9.7%) |
| Days to composite primary outcome, median [IQR] | 5 | 9.5 | 3 | 6 | 0.6 | 6 |
| | [3–8.5] | [4.5–10.8] | [2–8] | [2–10] | | [2–10] |
| *Throughout the hospital stay* | | | | | | |
| n (%) | 13 (7.6%) | 12 (18.5%) | 7 (14%) | 28 (22.4%) | 0.003 | 60 (14.5%) |
| Days to composite primary outcome, median [IQR] | 7 | 14 | 3 | 12 | 0.042 | 10 |
| | [3–10] | [10–33] | [2–8] | [5–23] | | [3–23] |

SoC: Standard of Care; CCS: corticosteroid; TCZ: tocilizumab; CPAP: Continuous Positive Airway Pressure; OTI: Oro-Tracheal Intubation.

[^]Statistical test performed: chi-square test of independence; Kruskal-Wallis test; Fisher's exact test.

[1] Data available for 378/412 (91.8%) participants.

other small observational studies showed lack of difference for mortality risk at 7 and 14 days from hospital admission by comparing moderate/severe to critical COVID-19 patients (67% of them receiving co-administration of TCZ and CCS) [43], and lack of influence of TCZ administered alone on 30-days mortality among severe [44] or critical patients [16, 45–47]. Overall, results from RCTs on TCZ effectiveness in COVID-19 patients seem conflicting possibly because of different study design and heterogeneity in clinical severity classification, whereas

**Table 4. Comparison between survivors and non-survivors for COVID-19-related death [a],[b].**

| | Total (n = 398)[b] | Survivors (n = 361) | Non-survivors (n = 37) | p-value |
|---|---|---|---|---|
| **Healthcare facility** | | | | |
| Policlinico Umberto I, Rome | 184 (46.2%) | 167 (46.3%) | 17 (45.9%) | >0.9 |
| S. Maria Goretti, Latina | 214 (53.8%) | 194 (53.7%) | 20 (54.1%) | |
| **Demographic characteristics** | | | | |
| Age, years, median [IQR] | 62 [51–75] | 61 [50–72] | 77 [69–83] | <0.001 |
| Female gender, n (%) | 172 (43.2%) | 156 (43.2%) | 16 (43.2%) | >0.9 |
| Male gender, n (%) | 226 (56.8%) | 205 (56.8%) | 21 (56.8%) | |
| **Co-morbidities** | | | | |
| Sum, mean (±SD) | 1.33 (±1.46) | 1.24 (±1.39) | 2.27 (±1.87) | <0.001 |
| Charlson index, mean (±SD) | 1.43 (±1.81) | 1.36 (±1.79) | 2.14 (±1.87) | 0.004 |
| Chronic cardiovascular diseases | 172 (43.2%) | 147 (40.7%) | 25 (67.6%) | 0.003 |
| Chronic pulmonary diseases | 59 (14.8%) | 51 (14.1%) | 8 (21.6%) | 0.3 |
| Chronic renal failure | 32 (8%) | 25 (6.9%) | 7 (18.9%) | 0.02 |
| Diabetes Mellitus | 73 (18.3%) | 63 (17.4%) | 10 (27%) | 0.2 |
| Cancer | 31 (7.8%) | 25 (6.9%) | 6 (16.2%) | 0.055 |
| Obesity (BMI>30) | 22 (5.5%) | 15 (4.2%) | 7 (18.9%) | 0.002 |
| Dyslipidaemia | 43 (10.8%) | 38 (10.5%) | 5 (13.5%) | 0.6 |
| Hypothyroidism | 28 (7%) | 23 (6.4%) | 5 (13.5%) | 0.2 |
| Neurodegenerative diseases | 23 (5.8%) | 17 (4.7%) | 6 (16.2%) | 0.013 |
| Autoimmune disease | 20 (5%) | 16 (4.4%) | 4 (10.8%) | 0.1 |
| Other disease | 27 (6.8%) | 26 (7.2%) | 1 (2.7%) | 0.5 |
| **Clinical presentation at baseline** | | | | |
| Symptoms | | | | |
| Days symptoms onset, median [IQR][^] | 7 [3–10] | 7 [3–10] | 6 [4–8.5] | 0.4 |
| Fever, n (%) | 298 (77%)[1] | 269 (76%)[2] | 29 (83%)[3] | 0.5 |
| Cough, n (%) | 198 (49.7%) | 178 (49.3%) | 20 (54%) | 0.7 |
| Dyspnoea, n (%) | 134 (34.8%)[4] | 112 (32%)[5] | 22 (65%)[6] | <0.001 |
| Diarrhoea, n (%) | 42 (10.9%)[7] | 38 (10.8%)[8] | 4 (11.4%)[3] | 0.8 |
| Myalgia, n (%) | 95 (24.5%)[1] | 90 (26%)[2] | 5 (14%)[3] | 0.2 |
| Loss of taste/smell, n (%) | 23 (5.8%) | 23 (6.4%) | 0 (0%) | 0.2 |
| Conjunctivitis, n (%) | 5 (1.3%)[9] | 4 (1.1%)[10] | 1 (3%)[11] | 0.4 |
| Laboratory findings | | | | |
| P/F, median [IQR] | 352 [286–414] | 359 [300–419] | 230 [159–307] | <0.001 |
| White blood cells, n/μl, median [IQR] | 5,600 [4,090–7,705][12] | 5,515 [4,090–7,678][13] | 5,765 [4,335–8,258][14] | 0.6 |
| Neutrophils, n/μl, median [IQR] | 3,680 [2,500–5,855][15] | 3,620 [2,515–5,755][16] | 5,050 [2,240–7,700] | 0.15 |
| Lymphocytes, n/μl, median [IQR] | 1,070 [740–1,550][12] | 1,080 [770–1,600][12] | 820 [500–1,330] | <0.001 |
| N/L Ratio, median [IQR] | 3.4 [2–6.1][15] | 3.3 [1.9–5.7][16] | 5.9 [3–14.3] | 0.001 |
| Ferritin, μg/L, median [IQR] | 417 [209–863][17] | 386 [205–754][18] | 1,302 [606–3,029][19] | <0.001 |
| LDH, U/L, median [IQR] | 256 [200–331][20] | 248 [195–313][21] | 355 [276–458][6] | <0.001 |
| D-dimer, μg FEU/ml [IQR] | 0.81 [0.45–1.55][22] | 0.75 [0.44–1.47][23] | 1.55 [1.01–3.22][16] | <0.001 |
| CRP, mg/dl, median [IQR] | 3 [1–8][24] | 2 [1–6][25] | 8 [4–22][6] | <0.001 |
| **Lowest respiratory function and support throughout the hospital stay** | | | | |
| P/F lowest, median [IQR] | 228 [152–366][26] | 264 [162–371][27] | 98 [70–120][6] | <0.001 |
| Ambient-air, n (%) | 132 (33.2%) | 132 (36.6%) | 0 (0%) | <0.001 |
| Venturi Mask, n (%) | 139 (34.9%) | 130 (36%) | 9 (24.3%) | |
| CPAP helmet, n (%) | 101 (25.4%) | 90 (24.9%) | 11 (29.7%) | |
| OTI, n (%) | 26 (6.5%) | 9 (2.5%) | 17 (46%) | |

*(Continued)*

**Table 4.** (Continued)

| | Total (n = 398)[b] | Survivors (n = 361) | Non-survivors (n = 37) | p-value |
|---|---|---|---|---|
| **Concomitant treatment, n (%)** | | | | |
| Lopinavir/Ritonavir | 202 (50.8%) | 182 (50.4%) | 20 (54%) | 0.8 |
| Hydroxychloroquine | 309 (77.6%) | 277 (76.7%) | 32 (86.5%) | 0.3 |
| Low-molecular weight heparin | 248 (62.3%) | 220 (61%) | 28 (75.7%) | 0.14 |
| Antibiotic administration, n (%) | 300 (75.4%) | 273 (75.6%) | 27 (73%) | 0.9 |
| **Treatment group, n (%)** | | | | |
| SoC | 166 (41.7%) | 159 (44.1%) | 7 (18.9%) | 0.006 |
| CCS | 59 (14.8%) | 53 (14.7%) | 6 (16.2%) | |
| TCZ | 50 (12.6%) | 46 (12.7%) | 4 (10.8%) | |
| TCZ + CCS | 123 (30.9%) | 103 (28.5%) | 20 (54.1%) | |
| **Days to TCZ and/or CCS administration** | | | | |
| Days to TCZ, median [IQR] | 3 [1–5] | 2.5 [1–5] | 3 [1–6] | 0.6 |
| Days to CCS, median [IQR] | 1 [0–4] | 0 [0–4] | 1 [0–4.2] | 0.3 |
| **Duration of hospitalization** | | | | |
| Days, median [IQR] | 19 [11–30] | 19 [11–30] | 14 [10–29] | 0.2 |

SoC: Standard of Care; CCS: corticosteroid; TCZ: tocilizumab; LDH: Lactate dehydrogenase; CRP: C-Reactive Protein; CPAP: Continuous Positive Airway Pressure;

OTI: Oro-Tracheal Intubation; N/L Neutrophil to Lymphocyte ratio.

[a] Statistical test performed: chi-square test of independence; Kruskal-Wallis test; Fisher's exact test.

[b] data on participants by excluding 14 non-COVID related deaths.

[^]Days from symptom onset: n = 347/398 (87.2%).

Data available for *n* participants (%):

[1] 387/398 (97.2%);

[2] 352/361 (97.5%);

[3] 35/37 (94.6%);

[4] 385/398 (96.7%);

[5] 351/361 (97.2%);

[6] 34/37 (91.2%);

[7] 386/398 (97%);

[8] 351/361 (97.2%);

[9] 383/398 (96.2%);

[10] 350/361 (97%);

[11] 33/37 (89.2%);

[12] 393/398 (98.7%);

[13] 356/361 (98.6%);

[14] 36/37 (97.3%);

[15] 391/398 (98.2%);

[16] 354/361 (98.1%);

[17] 241/398 (60.6%);

[18] 222/361 (61.5%);

[19] 19/37 (51.4%);

[20] 363/398 (91.2%);

[21] 328/361 (90.9%);

[22] 282/398 (70.9%);

[23] 253/361 (70.1%);

[24] 374/398 (94%);

[25] 340/361 (94.2%);

[26] 364/398 (91.5%);

[27] 330/361 (91.4%).

**Table 5. Multivariate analysis[1] composite primary and secondary outcomes at day-14 & day-28 from hospital admission among all population and in subgroups by disease severity.**

| Composite outcome (IMV or death) by treatment group | All Participants (n = 412) | | | | Participants with baseline PaO$_2$/FiO$_2$ < 300 (n = 111) | | | |
|---|---|---|---|---|---|---|---|---|
| | at 14 days from hospital admission | | at 28 days from hospital admission | | at 14 days from hospital admission | | at 28 days from hospital admission | |
| | HR (95%CI) | p-value | HR (95%CI) | p-value | HR (95%CI) | p-value | HR (95%CI) | p-value |
| SoC | 1 (Ref) | - | 1 (Ref) | - | 1 (Ref) | - | 1 (Ref) | |
| CCS | 0.41 (0.10–1.76) | 0.230 | 0.90 (0.23–3.44) | 0.873 | 0.77 (0.38–1.53) | 0.454 | 0.60 (0.33–1.10) | 0.096 |
| TCZ | 0.45 (0.36–0.57) | <0.001 | 0.58 (0.51–0.66) | <0.001 | 0.60 (0.59–0.60) | <0.001 | 0.51 (0.47–0.56) | <0.001 |
| TCZ+CCS | 0.57 (0.36–0.90) | 0.017 | 0.97 (0.58–1.62) | 0.908 | 0.52 (0.46–0.58) | <0.001 | 0.53 (0.47–0.59) | <0.001 |
| **Secondary outcome (COVID-related death) by treatment group** | **All Participants (n = 412)** | | | | **Participants with baseline PaO$_2$/FiO$_2$ < 300 (n = 111)** | | | |
| | at 14 days from hospital admission | | at 28 days from hospital admission | | at 14 days from hospital admission | | at 28 days from hospital admission | |
| | HR (95%CI) | p-value | HR (95%CI) | p-value | HR (95%CI) | p-value | HR (95%CI) | p-value |
| SoC | 1 (Ref) | - | 1 (Ref) | - | 1 (Ref) | - | 1 (Ref) | - |
| CCS | 0.52 (0.16–1.67) | 0.271 | 0.90 (0.26–3.13) | 0.865 | 1.27 (0.92–1.75) | 0.144 | 1.08 (1.02–1.15) | 0.006 |
| TCZ | 0.29 (0.17–0.50) | <0.001 | 0.33 (0.21–0.51) | <0.001 | 0.59 (0.39–0.89) | 0.011 | 0.53 (0.29–0.96) | 0.037 |
| TCZ+CCS | 0.30 (0.25–0.37) | <0.001 | 0.67 (0.43–1.04) | 0.077 | 0.32 (0.20–0.51) | <0.001 | 0.43 (0.22–0.82) | 0.010 |

[1] analysis adjusted for: age, gender, PO$_2$/FiO$_2$ ratio at baseline, CRP at baseline, Days from symptoms onset at the baseline, sum of comorbidities.

SoC: Standard of Care; CCS: corticosteroid; TCZ: tocilizumab.

retrospective studies tend to show a protective effect on outcomes. Furthermore it should be underlined that a comparison among observational studies appears difficult because of differences related to: 1) heterogeneity for disease classification at baseline; 2) heterogeneity for clinical severity of participants (with TCZ more prone to be given in more severe clinical cases in retrospective studies, as in our work); 3) TCZ timing, dosage, and way of administration not standardized; 4) low statistical power due to small number of participants; 5) lack of standardized treatment in the SoC groups; 6) lack of evaluation of corticosteroid effects on outcomes when included in the SoC.

In this retrospective cohort study we observed a lower risk for IMV or ICU-transfer or death (primary composite outcome) in both TCZ (HR 0.45, 95%CI: 0.36–0.57, p<0.001) and TCZ+CCS (HR 0.57, 95%CI: 0.36–0.90, p = 0.017) groups at 14-days, and only in TCZ group (HR 0.58, 0.51–0.66, p<0.001) at 28-days from hospital admission. By limiting the analysis for the primary composite outcome to participants with more severe disease at baseline, the protective effect remained at both time-points in TCZ and TCZ+CCS groups. The protective effect for COVID-related death (secondary outcome) was similar to what observed for the primary composite outcome, with lower hazard ratios in the whole study population. Because of the above-mentioned reasons, comparison of observational study results on TCZ efficacy is difficult: with this limitation in mind, our results seem in agreement with some studies [17, 32,

37, 48] but in contrast with others [41, 44, 46]. In general, it is important to highlight that the clinical management of COVID-19 hospitalized patients is complex and not standardized also for the respiratory support, with SoC being possibly different among clinical centres and over-time. In order to better standardize the clinical management of our patients during the first wave of the pandemic, a simplified flow-chart for internal use was established. The respiratory support we provided to patients hospitalized in Infectious Diseases wards of both participant hospitals was oxygen support through Venturi Mask or CPAP Helmet. At the time of this study the ICU-transfer was limited only to patients in need of IMV because of shortage of ICU-beds. In our clinical approach, for patients in need of a respiratory support higher than that provided through Venturi Mask, we used a strategy with prolonged assistance through CPAP helmets that possibly may have given more time to act to immunomodulatory drugs and then limiting the need of IMV.

By comparing survivors vs non-survivors, as expected, we found that baseline age, burden of comorbidities, dyspnoea at the presentation, $PaO_2/FiO_2$ ratio, lymphocytopenia and higher inflammatory biomarkers were all determinants of COVID-related death. Considering the nature of observational study, treating physicians have assigned more severe cases to the TCZ+CCS group in which a higher COVID-related mortality was observed.

Concerning the safety of tocilizumab, in our study we observed only 3 cases of severe neu-tropenia which recovered without infectious complication, and no differences for secondary infections among study groups were found. No safety concerns on tocilizumab have been reported by the majority of observational and RCT studies. Few observational studies reported TCZ safety concerns: 3 serious adverse events [36], and higher risk for bacterial superinfec-tions [16, 30, 37, 38]. Among RCTs, only one study showed severe neutropenia associated to TCZ (p = 0.002), but secondary infections were more frequent in the placebo group (p = 0.03) [19].

The administration of corticosteroid alone was not associated to our study outcomes. It is noteworthy that the co-administration of TCZ and CCS was protective for the study primary and secondary outcomes, with exception of 28-days primary outcome, possibly because of the presence of few not-COVID-related deaths occurred late during hospital stay. Moreover, dif-ferently from other authors [15, 49, 50], in our study the co-administration of corticosteroid was protective without improving tocilizumab outcomes effect in the whole population, but only among those with more severe respiratory failure at baseline. Therefore, it might be possi-ble that different strategies of corticosteroid administration might lead to different outcomes. Interestingly, the RECOVERY Trial, published on July 17, 2020 (after the last recruitment for this study), showed a 28-days mortality reduction in COVID-19 patients receiving a 10-days course of dexamethasone, but only among those taking oxygen [11] and, similarly to what we observed, no difference for outcomes was observed using different corticosteroid molecules [51]. In our study, participants of CCS group have possibly received the treatment although not in need of oxygen support; this might justify the lack of influence of corticosteroids on study outcomes. In published RCTs on TCZ effectiveness, corticosteroids have been adminis-tered in all but one RCTs [18], but no significant effects on outcomes have been identified. Moreover, corticosteroids have been administered to >80% of participants in three RCTs in both studies arms [20, 22, 23] but conflicting results on outcomes have been observed, possibly related to the different baseline disease severity between trials.

Anticoagulants are part of the SoC because thrombotic complications are frequent in COVID-19, more in severe cases and mainly in lungs [9, 10], apparently independent from ongoing thromboprophylaxis [52] and possibly as a consequence of a localized thrombotic process [53]. In our study, LWMH was included in the SoC, as prophylaxis or therapy accord-ing to D-dimer values, without differences by comparing COVID-19 survivors vs non-

survivors. Moreover, although we did not have recorded major thrombotic events, we cannot exclude that they occurred after ICU-transfer of patients.

The main limitation of this study are related to its observational nature that might have introduced biases, mainly for patient assignment to treatment groups, as well as to the relatively small number of participants. Another important point to consider might be the heterogeneity of the treatments in the SoC. Indeed, different agents used alone or in combination as SoC at the time of this study (i.e. hydroxychloroquine, azithromycin, lopinavir/ritonavir) have been abandoned because of lack of efficacy [13, 54–56], but with different timing in the clinical practice of participating clinical centres. Finally, we cannot exclude residual confounding because of unmeasured relevant covariates or reporting errors. However, we could rely on high-quality patient records that included the vast majority of routinely collected COVID-19 patient data and minimized the possibility of reporting errors.

Presently, although it is more than one year that SARS-CoV-2 infection became pandemic, there is a lack of a fully effective therapy for COVID-19 pneumonia. The only widely recognized as effective is the corticosteroid therapy among patients in need of oxygen support, although there are still unanswered questions related to its use [13, 57]. Concerning tocilizumab, although its use is safe, definitive evidences establishing its role in the management of COVID-19 patients are not-yet fully elucidated. Overall, large observational studies on TCZ effectiveness were early dismissed whereas small RCTs (with different study design) were over-interpreted leading to many unanswered questions that need to be addressed [58].

## Conclusion

According to the results of this study, we think that tocilizumab may play a useful role in COVID-19 pneumonia treatment, mainly in patients with higher inflammatory markers and more severe disease at the time of drug' administration. Moreover, we think that the co-administration of corticosteroid may be beneficial in COVID-19 patients with baseline severe respiratory failure.

## Supporting information

**S1 Table. Composite primary outcome at 14-days from hospital admission by treatment group excluding patients receiving remdesivir (11 patients) and according to the number of TCZ infusions (one or two).**
(DOCX)

**S2 Table. Observational studies on tocilizumab efficacy in COVID-19 patients.**
(DOCX)

**S3 Table. Randomized clinical trials on tocilizumab efficacy in COVID-19 patients.**
(DOCX)

**S1 Dataset.**
(CSV)

## Acknowledgments

We would like to thank all staff members of COVID-study groups of both hospitals involved in this study.

**COVID-19 Study Group, S. Maria Goretti hospital, Latina:** Miriam Lichtner, Cosmo Del Borgo, Raffaella Marocco, Valeria Belvisi, Tiziana Tieghi, Margherita De Masi, Paola Zuccalà, Paolo Fabietti, Angelo Vetica, Vito Sante Mercurio, Anna Carraro, Laura Fondaco, Blerta Kertusha, Alberico Parente, Giulia Mancarella, Andrea Gasperin, Davide Caianiello, Marco Perla,

Jessica Luchetti, Giulia Passariello, Ginevra Gargiulo, Emanuela Del Giudice, Riccardo Lubrano, Melania Garante, Maria Gioconda Zotti, Antonella Puorto, Marcello Ciuffreda, Antonella Sarni, Gabriella Monteforte, Rita Dal Piaz, Emanuela Viola, Carla Damiani, Antonietta Barone, Barbara Mantovani, Daniela Di Sanzo, Vincenzo Gentili, Massimo Carletti, Massimo Aiuti, Andrea Gallo, Piero Giuseppe Meliante, Salvatore Martellucci, Oliviero Riggio, Vincenzo Cardinale, Lorenzo Ridola, Maria Consiglia Bragazzi, Stefania Gioia, Rosanna Venere, Emiliano Valenzi, Camilla Graziosi, Niccolò Bina, Martina Fasolo, Silvano Ricci, Maria Teresa Gioacchini, Antonella Lucci, Luisella Corso, Daniela Tornese, Francesco Equitani, Carmine Cosentino, Antonella Melucci, Iavarone Carlo, Desirè Mancini, Frida Leonetti, Gaetano Leto, Camillo Gnessi, Giuseppe Pelle, Iannarelli Angelo, Mario Iozzino, Adriano Ascarelli, Cesare Ambrogi, Iacopo Carbone, Giuseppe Campagna, Roberto Cesareo, Francesca Marrocco, Giuseppe Straface, Clelia Di Pippo, Alessandra Mecozzi, Valentina Isgrò, Gabriella Bonanni, Sergio Parrocchia, Giuseppe Visconti, Giorgio Casati.

**COVID-19 Study Group, Policlinico Umberto 1st, Rome:** Claudio Maria Mastroianni, Vincenzo Vullo, Maria Rosa Ciardi, Gabriella d'Ettorre, Gianluca Russo, Camilla Ajassa, Claudia D'Agostino, Vito Trinchieri, Maria Rosaria Cuomo, Cristina Mastropietro, Andrea Brogi, Paola Guariglia, Laura Antonelli, Anna Paola Massetti, Caterina Fimiani, Ivano Mezzaroma, Mario Falciano, Martina Carnevalini, Alessandra Oliva, Giancarlo Ceccarelli, Francesco Le Foche, Giancarlo Iaiani, Cristiana Franchi, Maurizio De Angelis, Alessandro Russo, Alessandro Lazzaro, Federica Marincola, Luigi Celani, Eugenio Nelson Cavallari, Francesca Cancelli, Alessandro Bianchi, Marta Santori, Marco Rivano Capparuccia, Fiammetta Tamburrini, Vincenza Lorusso, Marco Ridolfi, Cecilia Tosato, Paolo Vassalini, Federica Alessi, Giulia Savelloni, Patrizia Pasculli, Guido Siccardi, Francesco Cogliati Dezza, Gregorio Recchia, Matteo Candy, Lorenzo Volpicelli, Alessia Cruciata, Gabriella De Girolamo, Riccardo Ficco, Francesco Romani, Serena Maria Carli, Vera Mauro, Valeria Filippi, Silvia Di Bari, Francesca Gavaruzzi, Ambrogio Curtolo, Raissa Aronica, Elena Casali.

## Author Contributions

**Conceptualization:** Gianluca Russo, Angelo Solimini, Vincenzo Vullo, Maria Rosa Ciardi, Claudio Maria Mastroianni, Miriam Lichtner.

**Data curation:** Gianluca Russo, Paola Zuccalà, Maria Antonella Zingaropoli, Anna Carraro, Patrizia Pasculli, Valentina Perri, Raffaella Marocco, Blerta Kertusha, Cosmo Del Borgo, Emanuela Del Giudice, Laura Fondaco, Tiziana Tieghi, Claudia D'Agostino, Alessandra Oliva, Maria Rosa Ciardi, Miriam Lichtner.

**Formal analysis:** Gianluca Russo, Angelo Solimini.

**Investigation:** Gianluca Russo, Paola Zuccalà, Maria Antonella Zingaropoli, Anna Carraro, Patrizia Pasculli, Valentina Perri, Raffaella Marocco, Blerta Kertusha, Cosmo Del Borgo, Emanuela Del Giudice, Laura Fondaco, Tiziana Tieghi, Claudia D'Agostino, Alessandra Oliva.

**Methodology:** Gianluca Russo, Angelo Solimini, Paola Zuccalà, Cosmo Del Borgo, Alessandra Oliva, Vincenzo Vullo, Maria Rosa Ciardi, Claudio Maria Mastroianni, Miriam Lichtner.

**Supervision:** Claudio Maria Mastroianni, Miriam Lichtner.

**Writing – original draft:** Gianluca Russo.

**Writing – review & editing:** Angelo Solimini, Vincenzo Vullo, Maria Rosa Ciardi, Claudio Maria Mastroianni, Miriam Lichtner.

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
