## [Decision Letter · Decision Letter 0]

1 Jul 2021

PONE-D-21-08792

Real-life use of tocilizumab with or without corticosteroid in hospitalized patients with moderate-to-severe COVID-19 pneumonia: a retrospective cohort study

PLOS ONE

Dear Dr. Russo,

Thank you for submitting your manuscript to PLOS ONE. After careful consideration, we feel that it has merit but does not fully meet PLOS ONE’s publication criteria as it currently stands. Therefore, we invite you to submit a revised version of the manuscript that addresses the points raised during the review process.

We look forward to receiving your revised manuscript.

Kind regards,

Francesco Di Gennaro

Academic Editor

PLOS ONE

Additional Editor Comments :

only some minor suggestions to improve your paper

Journal Requirements:

Reviewers' comments:

Reviewer's Responses to Questions

**Comments to the Author**

1. Is the manuscript technically sound, and do the data support the conclusions?

Reviewer #1: Yes

Reviewer #2: Yes

2. Has the statistical analysis been performed appropriately and rigorously? 

Reviewer #1: Yes

Reviewer #2: Yes

3. Have the authors made all data underlying the findings in their manuscript fully available?

Reviewer #1: Yes

Reviewer #2: No

4. Is the manuscript presented in an intelligible fashion and written in standard English?

Reviewer #1: Yes

Reviewer #2: Yes

5. Review Comments to the Author

Reviewer #1: This is a technically clean and sound paper with statistical analysis performed rigorously. Moreover, limitations are explained appropriately. The entire paper is presented in an intelligible fashion and clearly written.

Reviewer #2: Generally the manuscript was well written and easy to read.

The authors have reported a treatment option that could potentially be of significant benefit in a group of Covid patients.

I have only 2 comments:

1. The overall idea of demonstrating the benefit of TCZ in addition to other standard therapy, in a group of patients who are extremely diverse in terms of severity and nature of the illness is extremely difficult.

Although the authors did acknowledge this with a statement in the Discussion Pg 2- "It is important to highlight that clinical management of COVID-19 hospitalized patients is complex and not standardized also for the respiratory support" - the authors could elaborate more on how the clinical management was standardised with perhaps a hospital policy with the assistance of a standard algorith or flow chart that could pose some level of standardisation in the treatment.

2. In the limitation section the authors should include the small numbers that could limit the validity of the results.

6. PLOS authors have the option to publish the peer review history of their article (what does this mean?). If published, this will include your full peer review and any attached files.

Reviewer #1: No

Reviewer #2: No

---

## [Author Response · Author response to Decision Letter 0]

4 Jul 2021

Response to the reviewers’ comment:

First of all, we would like to thank the reviewers for their comment. Below, our reply to the points raised by the reviewers:

3. Have the authors made all data underlying the findings in their manuscript fully available?

Reviewer #1: Yes

Reviewer #2: No

Authors: We added a file with all data underlying our findings as supplementary material and we added the sentence “All relevant data are within the manuscript and its Supporting Information files”.

Reviewer#2:

1. The overall idea of demonstrating the benefit of TCZ in addition to other standard therapy, in a group of patients who are extremely diverse in terms of severity and nature of the illness is extremely difficult.

Although the authors did acknowledge this with a statement in the Discussion Pg 2- "It is important to highlight that clinical management of COVID-19 hospitalized patients is complex and not standardized also for the respiratory support" - the authors could elaborate more on how the clinical management was standardised with perhaps a hospital policy with the assistance of a standard algorith or flow chart that could pose some level of

standardisation in the treatment.

Authors: the sentence considered was not referred to our setting, but it was in general. Although that, we agree with the reviewer and we have modified the mentioned sentence and added a new one as follows: “In general, it is important to highlight that the clinical management of COVID-19 hospitalized patients is complex and not standardized also for the respiratory support, with SoC being possibly different among clinical centres and overtime. In order to better standardize the clinical management of our patients, a simplified flow-chart for internal use was established”.

2. In the limitation section the authors should include the small numbers that could limit the validity of the results.

Authors: the small number of patients has been added as study limitation as requested

---

## [Decision Letter · Decision Letter 1]

16 Jul 2021

PONE-D-21-08792R1

Real-life use of tocilizumab with or without corticosteroid in hospitalized patients with moderate-to-severe COVID-19 pneumonia: a retrospective cohort study

PLOS ONE

Dear Dr. Russo,

Thank you for submitting your manuscript to PLOS ONE. After careful consideration, we feel that it has merit but does not fully meet PLOS ONE’s publication criteria as it currently stands. Therefore, we invite you to submit a revised version of the manuscript that addresses the points raised during the review process.

Please address the issues and revise accordingly.

We look forward to receiving your revised manuscript.

Kind regards,

Academic Editor

PLOS ONE

Reviewers' comments:

Reviewer's Responses to Questions

**Comments to the Author**

1. If the authors have adequately addressed your comments raised in a previous round of review and you feel that this manuscript is now acceptable for publication, you may indicate that here to bypass the “Comments to the Author” section, enter your conflict of interest statement in the “Confidential to Editor” section, and submit your "Accept" recommendation.

Reviewer #1: All comments have been addressed

Reviewer #3: (No Response)

2. Is the manuscript technically sound, and do the data support the conclusions?

Reviewer #1: Yes

Reviewer #3: Yes

3. Has the statistical analysis been performed appropriately and rigorously? 

Reviewer #1: Yes

Reviewer #3: Yes

4. Have the authors made all data underlying the findings in their manuscript fully available?

Reviewer #1: Yes

Reviewer #3: Yes

5. Is the manuscript presented in an intelligible fashion and written in standard English?

Reviewer #1: Yes

Reviewer #3: Yes

6. Review Comments to the Author

Reviewer #1: (No Response)

Reviewer #3: the paper is interesting in its concept and being based on "real life experience', the data offers a different prospective about use of TCZ either alone or in conjunction with GCs, as the published results about this drug is controversial. however, this manuscript could benefit from some modifications to pinpoint its significant results.

Abstract:

In conclusion subsection, re-write it to be more concise and to the point, according to your results does use of TCZ alone was beneficial to patients or adding GCS was more beneficial?

introduction:

It is apparent that this study took place in Italy, please add sentence referring to death rate by age in Italy (if available).

in the first paragraph discussing cytokines, reference no. 8 reported low level of IL-6 in association with critical COVID-19, please add more rationale for optioning to use drug act against IL-6 receptors (i;e bolster what is written in references 6,9 and 10).

does the presence of more virus variants affects the efforts to establish standardized therapy? please clarify in introduction.

sentence linked to reference 11 is ambiguous, please clarify.

please add three-sentences paragraph about TCZ, the cornerstone of this study.

Method section:

Figure 1 needs more bolstering, it would be best if the flowchart contains original enrolled number, number of patients that were excluded and their exclusion criteria, eligible enrolled patients and their inclusion criteria, distribution of patients on groups with subdivision number of deaths in every group, number of patients discharged alive, number of patients who needed ICU.

In SOC, does this hospital/region/national guidelines, please clarify as a footnote.

the dose of TCZ is somewhat different from the one used in Rashad et al., 2021 and Albertini et al., 2021, does the different dosage affected the measured outcomes/survivability?

authors mentioned "the majority of patients received methylprednisolone, please add the exact number, and what about he remaining patients in the group, did they receive dexamethasone? please clarfiy.

Results:

group classification belongs to method section, please amend.

does the co-morbidity at admission affected the survival rate?

results in its current format are confusing, please reorder to pinpoint the most significant results of the study. start with epidemiological data (age, co-morbidities, time between appearance of signs and admission, signs at admission).

followed by baseline parameters, treatment groups and parameters, adverse effects and survival rate and end with multivariate COX hazard regression model.

no need to repeat drugs used in SOC protocol in results section.

this paragraph " intubation occurs more frequently in TCZ group, 2/3 with TCZ+ GCs), please clarify this result, which group exactly? and what number "26" stands for?

Discussion:

although there were conflicting results about usage of TCZ, it was not discussed extensively, please amend.

also, based on your results, which was more beneficial, TCZ alone, TCZ with GCs or SOC alone? please discuss

does different GCs used affected the results outcomes?

does co-morbidities affect outcome regardless of used therapeutic regimens

which co-morbidity benefit/ worsen by TCZ either alone or in conjunction with GCS?

Conclusion

it is preferable if conclusion does not contain references, please re-write this section and use "take home message" of your most significant results and recommendations based on your results

7. PLOS authors have the option to publish the peer review history of their article (what does this mean?). If published, this will include your full peer review and any attached files.

Reviewer #1: No

Reviewer #3: No

---

## [Author Response · Author response to Decision Letter 1]

19 Aug 2021

RESPONSE TO REVIEWER

(19th August 2021)

Reviewer #3: 

the paper is interesting in its concept and being based on "real life experience', the data offers a different prospective about use of TCZ either alone or in conjunction with GCs, as the published results about this drug is controversial. however, this manuscript could benefit from some modifications to pinpoint its significant results.

Abstract:

Q1. In conclusion subsection, re-write it to be more concise and to the point, according to your results does use of TCZ alone was beneficial to patients or adding GCS was more beneficial?

A1. Modified as requested

Introduction:

Q2. It is apparent that this study took place in Italy, please add sentence referring to death rate by age in Italy (if available)

A2. We thank the reviewer for the suggestion. Data (with the corresponding reference) have been changed

Q3. in the first paragraph discussing cytokines, reference no. 8 reported low level of IL-6 in association with critical COVID-19, please add more rationale for optioning to use drug act against IL-6 receptors (i;e bolster what is written in references 6,9 and 10). 

A3. Reference n.8 is a review comparing IL-6 serum level in severe COVID-19, hyperinflammatory ARDS, sepsis, and CRS in the setting of CAR-T cells Therapy. The paper shows that the peak of serum IL-6 in severe COVID-19 was lower than what observed in the other conditions known to be associated to high serum level of IL-6. Tocilizumab is already authorized for the treatment of the CRS in the setting of CAR-T cells Therapy. We thanks the reviewer for its suggestion and, in order to avoid misinterpretation, the sentence has been modified focusing on the lower peak of IL-6 in severe COVID-19 in comparison to the other clinical conditions that have been specified in the manuscript. 

Q4. does the presence of more virus variants affects the efforts to establish standardized therapy? please clarify in introduction.

A4. We thank the reviewer for the suggestion, but our paper did not take in account viral variants (and their potential impact on pathogenesis and treatment) because that is beyond the objective of our study.

Q5. sentence linked to reference 11 is ambiguous, please clarify.

A5. The sentence related to ref 11 is about the results of the RECOVERY trial in which the reduced mortality related to the corticosteroid therapy was evident only in patients in need of oxygen support (being possibly harmful in those receiving CCS and not in need of oxygen supply (RR 1.19))

Q6. please add three-sentences paragraph about TCZ, the cornerstone of this study.

A6. The sentences have been added as requested

Method section:

Q7. Figure 1 needs more bolstering, it would be best if the flowchart contains original enrolled number, number of patients that were excluded and their exclusion criteria, eligible enrolled patients and their inclusion criteria, distribution of patients on groups with subdivision number of deaths in every group, number of patients discharged alive, number of patients who needed ICU.

A7. We have modified the figure 1 taking in account the reviewer suggestions and we have also better specified inclusion criteria in the main manuscript

Q8. In SOC, does this hospital/region/national guidelines, please clarify as a footnote.

A8. SoC was based on advices (not properly guidelines) from a panel of experts of the Italian Society of Infectious and Tropical Diseases (SIMIT, Società Italiana di Malattie Infettive e Tropicali) and this has been added in the text

Q9. the dose of TCZ is somewhat different from the one used in Rashad et al., 2021 and Albertini et al., 2021, does the different dosage affected the measured outcomes/survivability?

A9. We agree with the reviewers about the heterogeneity of the dosage (and ways of administration: sc, iv) of TCZ used in different studies and we cannot answer properly to this question because, as in our study, no Therapeutic Drug Monitoring has been done as in all other published studies on TCZ use in COVID-19. Pharmacokinetic studies on TCZ are available only among few rheumatologic patients (and healthy controls), but not in COVID-19 patients. Furthermore, in order to be more precise on this aspect (and showing the heterogeneity among studies), in Table S2 which resumes observational studies on TCZ in COVID-19, the doses and ways of administration have been already specified as footnote of the table for each study: this aspect has been also already raised in the discussion section.

Q10. authors mentioned "the majority of patients received methylprednisolone, please add the exact number, and what about he remaining patients in the group, did they receive dexamethasone? please clarfiy.

A10. As suggested by the reviewer, the proportion of patients taking methylprednisone or other CCS (dexamethasone) has been specified in the manuscript.

Results:

Q11. group classification belongs to method section, please amend. 

A11. Modified as suggested.

Q12. does the co-morbidity at admission affected the survival rate?

A12. Yes, the coefficient of the sum of comorbidities was significant in the fully adjusted survival analysis at 14-days (HR 1.23 95%CI: 1.10-1.38, p<0.001) and 28-days (HR 1.20, 95%CI: 1.10-1.32, p<0.001). A specific sentence has been added in the results section. 

Q13. results in its current format are confusing, please reorder to pinpoint the most significant results of the study. Start with epidemiological data (age, co-morbidities, time between appearance of signs and admission, signs at admission) followed by baseline parameters, treatment groups and parameters, adverse effects and survival rate and end with multivariate COX hazard regression model.

A13. We thank the reviewer for the suggestion that we have followed by modifying the organization of the results as requested 

Q14. no need to repeat drugs used in SOC protocol in results section.

A14. Modified as suggested

Q15. this paragraph " intubation occurs more frequently in TCZ group, 2/3 with TCZ+ GCs), please clarify this result, which group exactly? and what number "26" stands for?

A15. We observed 26 cases oro-tracheal intubation: 17 in TCZ+CCS and 7 in TCZ group. The sentence has been modified as requested.

Discussion:

Q16. although there were conflicting results about usage of TCZ, it was not discussed extensively, please amend.

A16. We did not agree with this observation. Results of many different studies on TCZ use in COVID-19 (all resumed in Table S2 and S3) have been discussed, as well as factors hampering results comparison between and among studies. 

Q17. also, based on your results, which was more beneficial, TCZ alone, TCZ with GCs or SOC alone? please discuss does different GCs used affected the results outcomes?

A17. As already stated in the discussion section of the manuscript, the co-administration of corticosteroid was protective without improving tocilizumab outcomes effect, excepting in cases with more severe respiratory disease. This concept has been added in the discussion as requested. Moreover, as requested, it has been specified that the use of different corticosteroid molecules (methylprednisolone and dexamethasone) did not changed study outcomes 

Q18. does co-morbidities affect outcome regardless of used therapeutic regimens which co-morbidity benefit/ worsen by TCZ either alone or in conjunction with GCS?

A18. Although interesting, we did not test an hypothesis for the interaction between co-morbidities and treatment groups because it was beyond the study objectives. Moreover, we do not think that the number of patients in some of the groups is enough to perform this analysis

Conclusion

Q19. it is preferable if conclusion does not contain references, please re-write this section and use "take home message" of your most significant results and recommendations based on your results

A19. We thank the reviewer for the suggestion, and we have modified the text accordingly

---

## [Decision Letter · Decision Letter 2]

1 Sep 2021

Real-life use of tocilizumab with or without corticosteroid in hospitalized patients with moderate-to-severe COVID-19 pneumonia: a retrospective cohort study

PONE-D-21-08792R2

Dear Dr. Russo,

We’re pleased to inform you that your manuscript has been judged scientifically suitable for publication and will be formally accepted for publication once it meets all outstanding technical requirements.

Kind regards,

Academic Editor

PLOS ONE

Additional Editor Comments (optional):

Reviewers' comments:

Reviewer's Responses to Questions

**Comments to the Author**

1. If the authors have adequately addressed your comments raised in a previous round of review and you feel that this manuscript is now acceptable for publication, you may indicate that here to bypass the “Comments to the Author” section, enter your conflict of interest statement in the “Confidential to Editor” section, and submit your "Accept" recommendation.

Reviewer #1: All comments have been addressed

Reviewer #3: All comments have been addressed

2. Is the manuscript technically sound, and do the data support the conclusions?

Reviewer #1: Yes

Reviewer #3: Yes

3. Has the statistical analysis been performed appropriately and rigorously? 

Reviewer #1: Yes

Reviewer #3: Yes

4. Have the authors made all data underlying the findings in their manuscript fully available?

Reviewer #1: Yes

Reviewer #3: Yes

5. Is the manuscript presented in an intelligible fashion and written in standard English?

Reviewer #1: Yes

Reviewer #3: Yes

6. Review Comments to the Author

Reviewer #1: Authors have taken care of all comments and my recommendation is to accept this paper for publication

Reviewer #3: The authors have answered all the raised questions and addressed all the comments / suggestions precisely.

7. PLOS authors have the option to publish the peer review history of their article (what does this mean?). If published, this will include your full peer review and any attached files.

Reviewer #1: No

Reviewer #3: No

---

## [Editor Report · Acceptance letter]

2 Sep 2021

PONE-D-21-08792R2 

Real-life use of tocilizumab with or without corticosteroid in hospitalized patients with moderate-to-severe COVID-19 pneumonia: a retrospective cohort study 

Dear Dr. Russo:

I'm pleased to inform you that your manuscript has been deemed suitable for publication in PLOS ONE. Congratulations! Your manuscript is now with our production department. 

Kind regards, 

on behalf of

Dr. Robert Jeenchen Chen 

Academic Editor

PLOS ONE